# Physical Education Teachers’ Perception of Teaching Effectiveness Related to Gender and Center Location in the Community of Extremadura

**DOI:** 10.3390/ijerph20032199

**Published:** 2023-01-26

**Authors:** Jorge Rojo-Ramos, María Mendoza-Muñoz, Ángel Denche-Zamorano, Nicolás Contreras-Barraza, Santiago Gomez-Paniagua, Carmen Galán-Arroyo

**Affiliations:** 1Physical Activity for Education, Performance and Health (PAEPH) Research Group, Faculty of Sports Sciences, University of Extremadura, 10003 Cáceres, Spain; 2Research Group on Physical and Health Literacy and Health-Related Quality of Life (PHYQOL), Faculty of Sport Sciences, University of Extremadura, 10003 Cáceres, Spain; 3Promoting a Healthy Society Research Group (PHeSo), Faculty of Sport Sciences, University of Extremadura, 10003 Caceres, Spain; 4Facultad de Economía y Negocios, Universidad Andres Bello, Viña del Mar 2531015, Chile; 5BioẼrgon Research Group, Faculty of Sports Sciences, University of Extremadura, 10003 Cáceres, Spain

**Keywords:** physical education, teacher effectiveness, gender, education, self-assessment

## Abstract

Teacher effectiveness (TE) is defined as a set of strategies implemented by the teacher to ensure the multidimensional development of students. This effectiveness has usually been evaluated by students, department heads or parents, but not by one of the fundamental elements of the educational process, the teacher. In the current study, the aim is to understand the self-assessed teaching effectiveness of Physical Education (PE) teachers in the Community of Extremadura, finding differences according to the gender of the teacher and the location of the center. Significant differences were found in the scores of both the items and the dimensions that make up the questionnaire, regarding variables such as gender and location of the center. Similarly, we found worrying values regarding the use of technology and the evaluation of teachers and students, so these findings should serve to open new lines of action to develop measures to improve teaching effectiveness depending on the educational context. Therefore, this research is considered a first step in the analysis of the perceptions and needs of the teaching job from the point of view of a PE teacher in Extremadura.

## 1. Introduction

Education plays a fundamental role in the progress of societies and countries throughout the world, being the main factor studied since the middle of the last century [1]. The figure of the teacher stands as a relevant element that enables the qualitative development of the entire population [2] and, therefore, the constant evaluation and improvement of their professional work facilitates the achievement of the best results in terms of the quality of education and the future of the country [3].

Teacher evaluation is defined as a systematic and mandatory process focused on assessing their teaching ability [4], following three fundamental guidelines: (1) maintaining a balance between formative and summative purposes of their evaluation; (2) clarifying evaluation criteria and reflecting teachers’ performance through the use of various tools; and (3) involving internal and external evaluators [5]. Likewise, self-evaluation processes are embedded in the evaluation itself [6], making it possible to identify teaching competence and effectiveness, thus generating decisions and strategies aimed at self-improvement [7]. 

Furthermore, a great teacher will manage to simplify his or her teaching so that students can understand it [8], influencing their learning by overcoming constraints such as group size or financial aspects [9]. Teaching effectiveness is defined as those activities carried out by the teacher during the teaching process that promote students to develop aspects related to cognitive, affective and psychosocial domains [10]. Numerous previous research already pointed out its vital importance in the educational context, as effective teaching helps and promotes students’ learning [11,12]. Likewise, this teaching effectiveness and teacher competence level have a strong connection to curriculum implementation [13]. In this sense, self-evaluation, used in the assessment of teachers, allows teachers to assess the competence and efficacy of their teaching and make the necessary decisions to better themselves [7]. In addition, the benefits that other studies [14,15] have highlighted about such self-assessment are that it allows teachers to have a voice and control over their own development, to know the strengths and weaknesses of their practices, to focus on an improvement plan, thus encouraging the continuous development of teaching, and the ability to carry out such assessment when, where and as often as they wish. 

The subject of Physical Education (PE) aims at physical, mental, emotional, social and intellectual development through learning at school through physical activity [16]. Physical Education (PE) has great potential to achieve adherence to physical exercise throughout the life cycle. Considering the motivational elements of PE participants, it is essential to encourage participation in physical activity and promote healthy physical exercise among them [17]. Physical education teachers seem to have a key role in achieving curricular objectives, putting scientific and pedagogical knowledge of sport into practice, monitoring the progress of the sport movement and dealing with local actors, such as parents [18].

In this sense, numerous instruments have been developed that provide PE teachers with self-assessment tools [19,20], but these do not contain the measures of validity and reliability. However, the SETEQ-PE [21] provides PE teachers with a valid, reliable and easy-to-use self-assessment tool, having six thematic units that represent essential elements of effective teaching and provide a theoretical framework shared by teachers to examine the degree to which some features of core teaching practices are implemented.

Extremadura is one of the Autonomous Communities of Spain, which presents peculiar economic, territorial and population characteristics that have conditioned it to a slight socio-economic backwardness with respect to the rest of the Spanish regions, with the risk of poverty being 10 points above the national average (21.7% vs. 32.3%) [22]. There are large territorial differences, with urban areas concentrating most of the economic activity, employment, facilities and services, and therefore most of the population of Extremadura. Whereas, in some cases, the density of rural areas is less than 10 inhabitants per km^2^, and the population lacks services in terms of quality of life and social welfare and must move to urban or better developed and populated municipalities in order to find better socio-economic opportunities and have their needs for basic facilities, such as health and education services, met. Therefore, studying how Extremadurians, taking into account their socio-economic peculiarities, develop their professional work in different sectors, such as education, can be very significant.

Taking into account all of the above, and the importance of teaching effectiveness for the development of society in all areas, the aim of this study is to explore the self-assessed teaching effectiveness of PE teachers in the region of Extremadura (Spain) to determine the differences according to sex and location of the center, as well as to determine the reliability of the dimensions that make up the questionnaire. This analysis will allow the different administrations in charge of the education system to develop the lines of action adapted to the specific needs of teachers, considering their own characteristics and those of the centers where they carry out their professional activity. Thus, it was hypothesized that the self-assessed teaching effectiveness of Physical Education teachers in the region of Extremadura (Spain) is different according to gender and school location, and that the dimensions of the questionnaire used for the assessment of this purpose are reliable.

## 2. Materials and Methods

### 2.1. Participants

The sample consisted of 257 PE teachers from rural and urban public schools (Table 1) belonging to the Community of Extremadura (Spain), with an average teaching experience of 17 years (SD = 6.5) and an average age of 44 years (SD = 5.1). All of the participants were chosen using a convenience sampling method that was not based on probability [23].

### 2.2. Instruments

To obtain the sociodemographic characteristics of the sample, a brief questionnaire was prepared consisting of 6 questions: sex, studies completed, center environment, education, age and years of experience.

The Self-Evaluation of Teacher Effectiveness in Physical Education (SETEQ-PE) questionnaire [21] was used to ascertain the self-perceptions of teacher effectiveness in the evaluated sample. This scale is made up of 25 items grouped into 6 factors: (1) learning environment (5 items), related to the teacher’s ability to support individualized physical, cognitive and emotional student development in a pleasant and safe environment; (2) student and teacher assessment (5 items), which encompasses issues related to the evaluation methods used to assess the teaching process; (3) application of the content of physical education (4 items), which refers to the objectives that the teacher selects to teach, according to the goals to be achieved by the students; (4) use of technology (4 items), which refers to the use of video, voice recorder and computer by the teacher, and internet searching by the students; (5) teaching strategies (3 items), which assesses elements of the instructional framework according to the lesson objectives and student needs, such as teaching styles and formats or interaction patterns and organized ways for practice; and (6) lesson implementation (4 items), which includes the instructional actions that the teacher applies in order to maximize learning and to ensure the student’s physical and emotional safety. Each of the items is rated on a Likert scale ranging from 1 to 5, with 1 being “strongly disagree” and 5 “strongly agree”. The authors obtained an overall reliability coefficient, for the scale, of 0.87 (Cronbach’s Alpha), with these coefficients being higher than 0.70 in each of the dimensions that make up the questionnaire [21].

### 2.3. Procedure

To save costs and make the surveys easier to distribute, it was decided to use the Google Forms tool to create an e-questionnaire that included sociodemographic questions, as well as a questionnaire for evaluating and interpreting self-evaluations of teacher effectiveness in PE [24]. The data were collected between January and March 2022. 

The sample was obtained by searching the database of public educational centers in the Autonomous Community of Extremadura (Spain), available at (https://ciudadano.gobex.es/ciudadano-portlet/printpdf/pdf?typepdf=3443&idDirectorio=775 (accessed on 7 January 2022)), and selecting the contact information for schools and institutes where primary and secondary education were taught.

### 2.4. Statistical Analysis

The Statistical Package for Social Sciences (SPSS) version 24.0, IBM Corp., IBM SPSS Statistics for MAC OS, Armonk, NY, USA) was used to analyze the data gathered. Cronbach’s Alpha was used to calculate the instrument’s reliability for each of its dimensions.

The Kolmogorov–Smirnov test [25] was used to determine whether the variables satisfied the assumption of normality and it was determined that this assumption was not met; hence, nonparametric tests were applied. Additionally, the differences between the different items and factors of the questionnaire according to sex and center location were analyzed using the Mann–Whitney U test [26]. To conclude, Cronbach’s alpha was used to analyze the reliability of each dimension of the instrument.

## 3. Results

The descriptive statistics for all items as well as the differences evaluated according to gender and center location are shown in Table 2. In terms of gender, significant differences were found in the first three items of the questionnaire ("learning environment" dimension); item 9, all items belonging to the third and fifth dimensions, and in three of the four items that make up the fourth and sixth dimensions of the scale. Similarly, significant differences were obtained in all the items of the questionnaire when observing the variables of the location of the center, except in three items (items 17, 18 and 24).

Table 3 shows the scores obtained in the dimensions of the questionnaire according to gender and location of the center. If we look at the gender variable, all variables show significant differences except the second one, “student and teacher assessment”. In addition, the differences are notable in all dimensions when analyzing the location of the center, excluding the last one.

The Cronbach’s Alpha values were satisfactory for each of the scale factors (Table 4) as they were above 0.7 [27].

## 4. Discussion

Since teaching effectiveness is one of the most important factors for the success of teaching, this study arose to meet the need of understanding the current state of teaching self-evaluation regarding Physical Education teaching in the Community of Extremadura. To achieve this goal, the SETEQ-PE questionnaire was administered to check if these teachers had the ability to build a correct learning environment, evaluate their work adequately, apply subject contents correctly, include technologies to support their teaching, develop teaching strategies or instruct students in their sessions.

As for the results related to the learning environment, they show that the great majority of teachers individualize their teaching, while ensuring student safety. These results are in agreement with those obtained by Tulyakul and his colleagues [28], who examined whether there were differences in this area depending on the number of years of teaching experience. Therefore, the good results obtained in this study may be conditioned by the extensive experience of the teaching staff. These assertions have already been made by Omare and his team, who demonstrated that teachers with more experience are better able to adapt to new pedagogical trends [29]. If we focus on gender, male teachers have better scores than their female colleagues, which may be due to a greater influence of teaching experience on effectiveness in female teachers [30], as more years of experience were also obtained by male teachers. Regarding the location of the center, the results follow the line of Zheng and coworkers [31], which showed that urban teachers felt more efficacious than rural teachers. 

Regarding evaluation issues, teachers in Extremadura show average values, and it is surprising that most of them do not consider involving other teaching colleagues in their evaluation, although students do participate in it. In this sense, Almutairi and Shraid [32] noted that subjectivity, preference or prejudice, and competitiveness for advancement can compromise the accuracy of peer evaluation. In contrast, some research has already pointed out that teachers’ self-evaluation was similar to that carried out by their superiors [33]; although, it conflicts with the findings of other studies which show that teachers’ self-assessment leads to misinformation or overestimation [34]. Additionally, because students have a tendency to overrate their teachers, it is important to use caution when interpreting the evaluations they complete [33]. No significant differences were found in the second dimension when measuring the gender variable, despite the fact that previous research found differences according to the sex of the teachers when self-assessing their efficacy [35]. In concordance with a previous study, we have already indicated that teachers in urban schools perceive their teaching effectiveness and evaluation methods to be better than their peers in rural areas [36]. 

With respect to the dimension referring to the application of the contents of PE, educators perceive very good results in their self-assessment. This aspect may be due to the fact that most of them include novel contents spread on the Internet, which combine elements of dance, aerobics and physical fitness [37]. However, there is also a trend of change toward pedagogical models that seek problem solving in real situations [38], moving away from the execution of specific technical gestures [39]. The results regarding gender do not coincide with those found by Block and his colleagues, who did not identify a difference between teachers according to their gender [40]. However, other publications consider the teacher´s gender as a very relevant factor when it comes to applying the contents of PE and influencing the students who attend the sessions [41,42]. Regarding the location of the center, our results indicate higher scores in urban centers. The previous literature has already reported that students in rural schools experience a decrease in interest in physical activity as they grow [43], mainly due to the lower number of sports programs offered in these areas [44] or a lack of facilities, considered a facilitator of PA in the rural population [45], which makes it difficult to apply the contents of the subject.

The use of technology in physical education classes has been a topic of much study in recent years [46]. In this case, the teachers have obtained average scores on the items of this dimension. The integration of technology in the area of physical education has been highly related to the previous and continuous preparation of teachers in terms of information and communication technologies [47]. Additionally, including technology in educational centers is an imperative need due to its multiple possibilities and the great improvements it produces in learning [48]. Nevertheless, some teachers consider it impossible to implement due to the low investment of the available budget [49]. The findings in terms of gender variables are novel, as previous research had not found significant differences between sexes so far [50,51,52]. When observing the differences obtained in the location of the center, teachers in rural centers have better scores than their colleagues who work in the cities [53]. According to Howley and colleagues [54], teachers in rural locations exhibited more favorable attitudes toward technology. Moreover, the effectiveness of its integration is much higher in rural areas [55].

In terms of the teaching strategies dimension, the teachers of PE in Extremadura obtained the best self-assessments, differing from those previously obtained by various studies carried out in different countries. Şirinkan and Gündoğdu [56] exposed that Turkish PE instructors mostly employed command and practice approaches in their courses, as well as teachers in northern European countries [57]. Therefore, it has been recommended that specific teaching strategies be developed for each of the content areas of PE [58] and that the learning outcomes of the course be aligned with these strategies [59]. According to the research and earlier studies, female teachers believe the command style to be the most productive for their students’ learning [60]. Moreover, urban teachers reported higher scores than their rural peers. These observations were already reported by Jovanović and Minić [61], who found a smaller range of strategies in rural teachers.

Lastly, the values obtained in the last dimension of the questionnaire are excellent. However, there are several factors that limit the implementation of the educational plans, such as teacher inexperience, lack of professionalism, clothing, and resource and space limitations [62]. The previous literature did not find any gender differences in the implementation of key competencies during PE classes [63,64], in contrast to this work. Likewise, no differences were found when examining the variable location of the center.

### Limitations and Futures Research

This study is one of the first to describe self-assessed teaching efficacy in physical education teachers in the region of Extremadura (Spain), identifying differences according to gender and school location. While this study provides important information on self-assessed teaching effectiveness, some limitations should be considered when assessing the generalizability of these findings. This study used a convenience sample of participants who were more likely to be middle-aged and highly experienced teachers. In this sense, the average experience of the teachers could overestimate the results, since it has been found to be highly related to teaching effectiveness [65,66]. Therefore, selection bias may influence the results; thus, we should be cautious when citing these results for generalizations about self-assessed teaching effectiveness in physical education teachers. 

Consequently, future studies should expand the sample of analysis in the community, as well as group this sample according to the different educational levels in order to understand the current state of teaching in PE at a global level, developing specific measures and actions that can take into account all the conditioning factors (e.g.: level of education where the work is carried out, center environment, material resources available, type of students, type of educational content…). It would also be useful to include younger teachers in the analyses so that the higher educational needs of future PE teachers can be sensed once they have had the opportunity to put their university training into practice, as well as other professionals involved in PE classes.

## 5. Conclusions

This study indicates that Physical Education teachers in the Community of Extremadura have a positive self-perception of their teaching effectiveness in four of the six dimensions that make up the questionnaire. In general, males perceived their teaching effectiveness and assessment methods better than females, as well as teachers in urban versus rural areas. These results indicate the training and facilitation needs for the implementation of two methods, namely teacher and student evaluation and the use of technology. However, the scores obtained in the dimensions “Student/Teacher Assessment” and “Use of Technology” were slightly lower than the rest for all study subgroups; therefore, it might be necessary to establish measures and initial and continuous training courses that allow the application of new evaluation instruments and specific contents of PE that involve new technologies.

In this sense, the evaluation of teaching effectiveness is essential for the entire teaching process since students’ learning and development depend on them. In this case, it is even more important because of the possibilities of PE as a means of generating healthy life habits and physical activity, commitment to the environment, cooperative activities, problem solving or nutritional aspects.

## Figures and Tables

**Table 1 ijerph-20-02199-t001:** Frequency distribution of the sample (*n* = 257).

Variables	Categories	*n*	%
Gender	Male	126	49.0
	Female	131	51.0
Studies Completed	Teacher Training PE	77	30.0
	Physical Activity and Sport Sciences	118	45.9
	Both	62	24.1
Teaching	Primary School	113	44.0
	Secondary/High School	144	56.0
Center Environment	Rural	113	44.0
	Urban	144	56.0
		**Mean**	**SD**
Teaching experience (years)	Male	17.75	5.45
	Female	16.77	7.28
Age	Male	43.46	3.85
	Female	44.17	6.18

*n* = sample number; PE = Physical Education; SD = standard deviation.

**Table 2 ijerph-20-02199-t002:** Descriptive statistics and differences in SETEQ-PE questionnaire items based on gender and center location.

		Gender			Center location		
Item	Total	Female	Male		Rural	Urban	
	M (SD)	M (SD)	M (SD)	*p*	M (SD)	M (SD)	*p*
Learning environment							
1. Do you individualize your teaching so that each of your students improves emotionally and socially?	4.09 (0.81)	3.82 (0.81)	4.37 (0.71)	<0.01 **	3.73 (0.66)	4.37 (0.82)	<0.01 **
2. Do you individualize your teaching so that each of your students improves kinetically?	4.00 (0.66)	3.82 (0.42)	4.19 (0.79)	<0.01 **	3.80 (0.63)	4.17 (0.64)	<0.01 **
3. Do you individualize your teaching so that each of your students improves cognitively?	4.19 (0.65)	4.00 (0.71)	4.39 (0.51)	<0.01 **	3.90 (0.61)	4.42 (0.59)	<0.01 **
4. Is student safety (physical, emotional, social) guaranteed during your lesson?	4.49 (0.70)	4.50 (0.63)	4.48 (0.78)	0.59	4.19 (0.73)	4.72 (0.59)	<0.01 **
5. Do you modify your lesson plan to ensure motivation, progress, and safety of students?	4.58 (0.51)	4.55 (0.52)	4.60 (0.51	0.39	4.48 (0.52)	4.65 (0.49)	<0.01 **
Student and teacher assessment							
6. Do students participate in the evaluation of your teaching (e.g., with a questionnaire)?	2.85 (1.05)	2.92 (0.99)	2.78 (1.12)	0.40	2.42 (1.04)	3.18 (0.94)	<0.01 **
7. Do you involve your students in the evaluation of their classmates?	3.07 (1.14)	3.16 (0.88)	2.97 (1.36)	0.26	2.49 (1.05)	3.52 (1.00)	<0.01 **
8. Do you invite your colleagues to evaluate your teaching?	1.96 (0.90)	2.02 (0.89)	1.90 (0.92)	0.27	1.65 (0.87)	2.21 (0.86)	<0.01 **
9. Do you use techniques to evaluate students cognitively and socially (e.g., multiple choice questions, rubrics)?	3.34 (1.12)	3.52 (1.11)	3.16 (1.11)	0.02 *	2.92 (1.18)	3.67 (0.95)	<0.01 **
10. Do you use other techniques (e.g., evaluation during game, evaluation scales, and rubrics) for the motor evaluation of students?	4.07 (0.96)	4.19 (0.92)	3.95 (0.98)	0.05	3.72 (0.93)	4.35 (0.88)	<0.01 **
Application of the content of PE							
11. Do you teach tactics, rules, and regulations of educational and sport games?	4.70 (0.47)	4.56 (0.50)	4.85 (0.38)	<0.01 **	4.62 (0.49)	4.77 (0.44)	<0.01 **
12. Do you integrate issues like nutrition, obesity, smoking, drugs, and tactics in your teaching?	4.33 (0.68)	4. 23 (0.46)	4.44 (0.83)	<0.01 **	4.13 (0.34)	4.49 (0.82)	<0.01 **
13. Do your students acquire knowledge and skills from other subjects (e.g., Language, Mathematics, Geography, and History) through your lesson?	3.81 (1.04)	3.42 (0.84)	4.21 (1.08)	<0.01 **	3.58 (1.00)	3.98 (1.03)	<0.01 **
14. Do you teach techniques (e.g., of skills, physical fitness, etc.)?	4.65 (0.48)	4.56 (0.50)	4.75 (0.44)	<0.01 **	4.49 (0.50)	4.78 (0.41)	<0.01 **
Use of technology							
15. Do you use videos for teaching?	4.02 (0.86)	3.78 (0.84)	4.27 (0.82)	<0.01 **	4.25 (0.80)	3.84 (0.87)	<0.01 **
16. Do you make use of the computer to teach?	3.53 (1.04)	3.37 (1.08)	3.69 (0.99)	0.02 *	3.75 (1.06)	3.35 (1.00)	<0.01 **
17. Do you assign tasks that require students to search for information on the Internet?	2.77 (0.89)	2.53 (0.89)	3.03 (0.82)	<0.01 **	2.81 (0.88)	2.74 (0.90)	0.56
18. Do you use a video and voice recorder to evaluate your teaching?	1.86 (0.83)	1.78 (0.84)	1.94 (0.83)	0.10	1.83 (0.84)	1.88 (0.83)	0.65
Teaching strategies							
19. Do you employ student-centered teaching styles (e.g., exploration, problem solving, etc.) according to learning objectives and student needs?	4.12 (0.80)	3.89 (0.79)	4.37 (0.74)	<0.01 **	3.85 (0.60)	4.34 (0.87)	<0.01 **
20. Apart from partial and whole practice, do you employ methods of group/random, constant/varying practice?	4.12 (0.95)	3.94 (1.12)	4.30 (0.70)	0.04 *	3.87 (1.06)	4.31 (0.80)	<0.01 **
21. Do you use a wide variety of media (e.g., tables, posters, music, cards)?	3.75 (0.92)	3.35 (0.85)	4.16 (0.81)	<0.01 **	3.35 (0.88)	4.06 (0.84)	<0.01 **
22. Do you inform your students about what they are going to learn?	4.68 (0.50)	4.60 (0.54)	4.75 (0.47)	0.01 *	4.90 (0.33)	4.50 (0.56)	<0.01 **
Lesson Implementation							
23. Does your teaching plan involve objectives and specific movement, cognitive, and social goals for each class?	4.60 (0.67)	4.64 (0.60)	4.55 (0.73)	0.54	4.74 (0.50)	4.48 (0.76)	<0.01 **
24. Do you have a teaching plan for each lesson?	4.66 (0.67)	4.59 (0.64)	4.73 (0.70)	<0.01 **	4.66 (0.62)	4.65 (0.71)	0.61
25. Do you demonstrate objectives to be learned, when it is required by the course?	4.54 (0.68)	4.47 (0.66)	4.61 (0.70)	0.02 *	4.38 (0.67)	4.67 (0.67)	<0.01 **

M = mean; SD =standard deviation; The correlation is significant at the ** *p* < 0.01; * *p* < 0.05.

**Table 3 ijerph-20-02199-t003:** Differences in SETEQ-PE dimensions according to gender and center location.

		Gender			Center location		
Dimensions	M (SD)	Female	Male	*p*	Rural	Urban	*p*
Learning Environment	4.27 (0.53)	4.14 (0.51)	4.40 (0.50)	<0.01 **	4.02 (0.43)	4.46 (0.51)	<0.01 **
Student/Teacher Assessment	3.06 (0.83)	3.16 (0.74)	2.95 (0.91)	0.11	2.64 (0.79)	3.38 (0.70)	<0.01 **
Application of the content of PE	4.38 (0.58)	4.19 (0.49)	4.56 (0.59)	<0.01 **	4.20 (0.47)	4.50 (0.62)	<0.01 **
Use of Technology	3.04 (0.75)	2.86 (0.79)	3.23 (0.67)	<0.01 **	3.16 (0.73)	2.95 (0.76)	0.03 *
Teaching Strategies	4.06 (0.69)	3.81 (0.66)	4.31 (0.63)	<0.01 **	3.78 (0.50)	4.28 (0.73)	<0.01 **
Lesson Implementation	4.62 (0.51)	4.58 (0.43)	4.66 (0.58)	<0.01 **	4.67 (0.40)	4.57 (0.58)	0.77

The correlation is significant at the ** *p* < 0.01; * *p* < 0.05.

**Table 4 ijerph-20-02199-t004:** Reliability values of the dimensions of the questionnaire.

Item	Cronbach’s Alpha
Learning Environment	0.84
Student/Teacher Assessment	0.86
Application of the content of PE	0.84
Use of Technology	0.85
Teaching Strategies	0.75
Lesson Implementation	0.81

## Data Availability

The datasets used during the current study are available from the corresponding author on reasonable request.

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
