# Peer review of "Physical Education Teachers’ Perception of Teaching Effectiveness Related to Gender and Center Location in the Community of Extremadura"

_ijerph, 2023, doi:10.3390/ijerph20032199_

Round 1

Author Response

The answers to your suggestions can be found in the following Word document

Reviewer 2 Report

It is a good and interesting paper. I justify my opinion by the following citation: Teaching effectiveness is defined as those activities carried out by the teacher during the teaching process that promote students to develop aspects related to cognitive, affective and psychosocial domains.

It it is very rare to find research that combine these three domains, including affective domain.

The methodological procedures are adequate and are very well explained. In this sense, the discussion are clear and interesting to the readers.

The authors were careful to still present the limitations of the study and gives some clues for further studies.

Just the conclusions can be improved but it is not required. It is only an suggestion. 

Author Response

(The authors gave the same response as above.)

Reviewer 3 Report

Thanks for inviting me to review this manuscript. I think this paper addresses an interesting and important topic, however, there are some issues to be solved before it can be published.

Introduction:

1. I found the introduction could be framed more clearly regarding the relationship between teaching evaluation and self-assessment. Are they all conducted by the teacher themselves? Why self-assessment is important to be studied?

2. You specifically focused on the region of Extremadura, is there any reasons? Justifications regarding why this region should be provided. 

3. What are your research hypotheses? research hypothese should be formulated before you do any testing, given this is a quantitative study.

Method/results:

1. I wonder why this Mann–Whitney U test was adopted to examine difference not t-test? 

2. what is the effect size?

Discussion:

1. how is your results similar or difference when compared with previous research?

2. in the limitation, common method bias should be mentioned. 

Author Response

(The authors gave the same response as above.)
